# Correlation Analysis of Sperm Cryopreservation Quality with Serum Testosterone and Sperm gDNA Methylation Levels in Xiaoshan Chickens

**DOI:** 10.3390/ani15121745

**Published:** 2025-06-13

**Authors:** Xue Du, Duoxi Li, Luya Jia, Xiaopeng Tong, Zhiyuan Huang, Yali Liu, Panlin Wang, Ayong Zhao

**Affiliations:** College of Animal Science and Technology & College of Veterinary Medicine, Zhejiang A&F University, Hangzhou 311300, China; duxue@zafu.edu.cn (X.D.); ldx2327@163.com (D.L.); 19730560159@163.com (L.J.); zaoyooooo@163.com (X.T.); 15275237629@163.com (Z.H.); 19730568002@163.com (Y.L.); wangpl2021@zafu.edu.cn (P.W.)

**Keywords:** Xiaoshan chicken, semen cryopreservation, DNA methylation, testosterone

## Abstract

Semen cryopreservation refers to the method of long-term storage of sperm through ultra-low temperature technology, aiming to maintain the biological activity and fertilization ability of the sperm for subsequent use. Chicken semen cryopreservation is crucial for utilizing high-quality genetics, but the outcomes are limited. This study explored semen cryopreservation outcomes with the perspective of serum indicators and sperm DNA epigenetic modification. Through serum detection, semen quality testing, WGBS (whole-genome bisulfite sequencing), and BSP (bisulfite sequencing PCR), a significant correlation between serum testosterone levels and semen cryopreservation quality was established, and key genes linked to cryopreserved sperm motility were sequenced and verified as well. These findings provide insights into improving semen preservation and have potential applications in poultry breeding and cryopreservation technologies.

## 1. Introduction

Semen cryopreservation represents a significant breakthrough in artificial insemination (AI), accelerating breeding and genetic improvement in livestock and poultry. It has facilitated the exchange of superior genetic material across regions, extended the usable lifespan of valuable semen, and contributed significantly to the development of the livestock industry [1]. However, despite its success in bovine species, semen cryopreservation in poultry, particularly chickens, remains in its infancy due to the high susceptibility of chicken sperm to cryoinjury.

Compared to bovine semen, the effectiveness of cryopreserving chicken semen is considerably lower, with sperm motility post thaw averaging around 25% [1]. Fertilization rates in cryopreserved chicken semen range widely from 0% to 83.3% [2,3,4,5,6], though most studies report rates between 50% and 70% [6]. Several biological factors contribute to these challenges, such as susceptibility to oxidative stress and low membrane stability. The membranes of avian sperm, including those of chickens, contain a higher proportion of polyunsaturated fatty acids compared to mammalian sperm. This composition makes them more prone to oxidative stress during freezing [7]. The cholesterol-to-phospholipid ratio in chicken sperm is extremely low, further weakening sperm membrane stability during cryopreservation [8]. Consequently, chicken sperm experience significant losses in membrane integrity, vitality, and fertilization potential during the freezing and thawing process [9].

Increased reactive oxygen species (ROS) production during semen cryopreservation can affect spermatozoon structure and function, as well as DNA methylation levels [10]. Research on the effects of cryopreservation has revealed changes in sperm DNA methylation across various species. For instance, in horses, the DNA methylation level increased significantly post thawing (from 0.6% to 6.4%, *p* < 0.01) [11], and a similar trend was observed in sheep [12]. Sperm DNA methylation is associated with chromatin integrity, DNA quality, and sperm functionality. In humans, sperm DNA methylation levels have been both positively and negatively correlated with sperm motility, suggesting that methylation may affect different sperm characteristics in complex ways [11,13,14]. While studies on horses found no significant correlation between DNA methylation and sperm motility, morphology, or membrane integrity [15], the relationship between sperm DNA methylation and sperm cryopreservation quality in chickens remains underexplored and requires systematic investigation.

Testosterone plays a vital role in spermatogenesis, with a minimum threshold required for normal testicular function. In clinical settings, serum testosterone levels are often used as a key endocrine indicator for assessing male infertility [13]. Research has shown that testosterone levels are linked to sperm morphology and quality across species. For instance, in dolphins, serum testosterone is positively correlated with midpiece length but negatively with sperm head width and total sperm length [14]. Similarly, in mice, dietary supplementation with D-aspartic acid significantly improved sperm quality and in vitro fertilization rates, which were positively correlated with increased serum luteinizing hormone and testosterone levels [16]. However, in chickens, the relationship between testosterone and sperm motility appears less straightforward. For example, studies on Beijing-You chickens found that while testosterone levels were associated with sperm production, they did not correlate with sperm motility [17]. Investigating the relationship between testosterone levels and the quality of cryopreserved sperm in chickens could help optimize semen collection and training practices, reduce costs, and improve breeding efficiency.

The Xiaoshan Chicken, a valuable breed in China, is known for its robust physique and high meat quality, making it a valuable strain for high-quality broiler breeding. However, the current effectiveness of cryopreservation for Xiaoshan Chicken semen does not yet meet industry standards. This study aims to (1) investigate the quality of cryopreserved Xiaoshan Chicken sperm and identify factors affecting its viability; (2) establish relationships between sperm quality and serum hormone levels (e.g., testosterone) and biochemical indicators, potentially enabling the selection of high-quality cockerels for semen cryopreservation based on serum analysis; and (3) examine the impact of sperm DNA methylation levels on semen cryopreservation quality to gain insights into the epigenetic modifications associated with semen preservation.

By identifying key indicators that influence semen cryopreservation success in Xiaoshan Chickens, this research could inform the development of more effective cryopreservation methods and optimize the selection of breeding cockerels. Moreover, understanding the role of DNA methylation in sperm functionality may lead to the creation of improved cryopreservation reagents, enhancing overall reproductive efficiency in poultry breeding programs.

## 2. Materials and Methods

### 2.1. Sampling

The Xiaoshan Chickens were cultured in Hangzhou Qiantang Donghai Breeding Limited Liability Company, Hangzhou, China (we have obtained written informed consent from the company). A total of 200 healthy 180-day-old male Xiaoshan Chickens, with uniform body weight and well-defined secondary sexual characteristics, were randomly selected for individual cage rearing with free access to food and water. Semen collection was conducted following the method described by Du et al. [18]. Semen samples free from fecal and blood contamination were collected for cryopreservation. To ensure quality and motility, only fresh semen samples with motility greater than 70% and sperm concentration of approximately 200–300 million/mL were used for cryopreservation. The cryopreservation protocol was as follows: Semen was collected via a two-person dorsal abdominal massage method and immediately mixed with an equal volume of diluent (improved Lake’s diluent containing fructose, potassium citrate tribasic monohydrate, L-sodium glutamate, magnesium acetate tetrahydrate, sodium acetate anhydrous, and gentamycin sulfate). The mixture was incubated at 4 °C for 30 min. Subsequently, glycerol cryoprotectant (improved Lake’s diluent containing 12% (*vol*/*vol*) glycerol) was added at a ratio of two volumes of cryoprotectant to one volume of semen. The mixture was then thoroughly homogenized and allowed to equilibrate at 4 °C for an additional 30 min. The processed semen was then loaded into labeled 0.25 mL straws (IMV Technologies, L’Aigle, France), sealed with sealing powder, and packed into storage bags. The straws were placed 5 cm above the liquid nitrogen and vaporized with liquid nitrogen for 5 min, and the semen was lowered from 4 °C to −35 °C and then immersed in a liquid nitrogen tank for long-term preservation.

Blood samples were collected from the wing veins of 102 Xiaoshan Cockerels. A 2 mL aliquot of blood was placed into pro-coagulant tubes (20220510, Saihua, Shandong, China) and centrifuged at 1000× *g* for 5 min after coagulation. The serum was then collected and stored at −80 °C for further measurement.

All procedures were conducted following the *Guide for the Care and Use of Laboratory Animals: Eighth Edition*, ISBN-10: 0-309-15396-4. The animal protocol was approved by the Ethics Committee for Animal Experiments of Zhejiang A&F University.

### 2.2. Semen Thawing and Sperm Quality Assessment

Semen thawing was performed following the method described by Du et al. [18]. The detailed procedure is as follows: Frozen semen straws were retrieved from the liquid nitrogen tank and placed in a 37 °C water bath for 40 s to thaw. The straws were then removed, and the surface moisture was wiped off. Both ends of the straws were trimmed using scissors to remove the sealing powder and cotton plug, allowing the semen to flow naturally into a pre-prepared 1.5 mL centrifuge tube. A diluent was added to adjust the sperm concentration to 3 × 10^7^ cells/mL. Sperm qualities were measured using a Computer-Assisted Sperm Analysis system (CASA, 024905, CEROS II™, Hamilton Thorne, Beverly, MA, USA). Based on the percentage of motile sperm in a straight-line trajectory (LIN) and the total motile sperm ratio (TM), the samples were divided into an H group (LIN > 25% and TM > 50%) and L group (LIN < 5% and TM < 20%).

### 2.3. Serum Indicator Measurement

The concentrations of testosterone (T), estradiol (E_2_), superoxide dismutase (SOD), and glutathione peroxidase (GSH-px) in chicken serum were measured using an enzyme-linked immunosorbent assay (ELISA). The procedures were carried out according to the manufacturer’s instructions provided with the ELISA kits, which were supplied by Shanghai Kexiabo Biotechnology Co., Ltd., Shanghai, China.

### 2.4. gDNA Extraction

Genomic DNA (gDNA) was extracted using a DNA extraction kit (9765, TaKaRa MiniBEST Universal Genomic DNA Extraction Kit, Takara, Beijing, China). Its quality and concentration was subsequently measured.

### 2.5. Library Construction and Quality Control

Library construction and quality control were performed following the methods described by Krueger et al. [19] and Langmead et al. [20]. The detailed procedure was as follows: 100~200 ng of gDNA was sonicated and subjected to bisulfite treatment after the addition of λDNA. An adenine (A) base was added to the 3′ end, and adapter sequences were introduced at the 5′ end. The resulting product was used as a template for PCR amplification. The quality of the amplified library was assessed using an Agilent 2100 Bioanalyzer (Agilent Technologies Inc., Santa Clara, CA, USA) and an Agilent High Sensitivity DNA Kit (5067-4626, Agilent Technologies Inc., Santa Clara, CA, USA). The total concentration of the library was measured using PicoGreen (E6090, Quantifluor-ST fluorometer, Promega, Madison, WI, USA; P7589, Quant-iT PicoGreen dsDNA Assay Kit, Invitrogen, Carlsbad, CA, USA). After the normalization of the multiplexed DNA libraries, they were pooled in equal volumes. The pooled DNA libraries were then diluted stepwise and quantified before being sequenced in PE150 mode using the Illumina Novaseq 6000 sequencing platform (Illumina, San Diego, CA, USA).

### 2.6. Data Analysis

#### 2.6.1. Analysis of Sperm Quality and Serum Indicators

Using SPSS 27.0 software, the quality parameters of cryopreserved sperm and serum indicators were analyzed. The analysis results are presented as mean ± standard deviation and an independent-samples *t*-test was employed. A *p*-value ≤ 0.05 indicates significant differences between groups. Additionally, Pearson correlation analysis was used to assess the relationship between two variables, with the correlation coefficient r reported. A *p*-value ≤ 0.05 denotes a significant correlation, while a *p*-value ≤ 0.01 denotes a highly significant correlation.

#### 2.6.2. Methylation Sequencing Data Analysis

Trimmomatic software version 0.36 [21] was used to filter the connectors and low-quality reads to obtain clean data for further analysis.

Reads Mapping: The reference genome (Gallus gallus, GRCg6a, https://asia.ensembl.org/Gallus_gallus (accessed on 19 August 2024)) was downloaded from a genome website. The filtered reads were mapping to the reference genome using bismark bowtie2 (https://www.bioinformatics.babraham.ac.uk/projects/bismark/ (accessed on 19 August 2024)).

Methylation Statistics: By statistical bismark results, the methylation context (CpG, CHH, CHG) information in all reads aligned to the genome was obtained. For the identified methylation sites, the number of mCs and non-mCs at each site and the degree of methylation were calculated.

Differentially methylated region (DMR) analysis: DMR analysis was performed using DSS analysis software according to the spatial correlation, read depth of the sites and variance among biological replicates.

Differentially methylated gene (DMG) annotation: Based on the distribution of DMRs on the genome, the gene body region (from TSS to TES) that overlapped with DMRs was enriched, as well as the genes whose promoter retions (2K up stream of the transcription start site) overlapped with DMRs. All these genes were DMGs.

KEGG enrichment analysis: ClusterProfiler software (version 3.16.1) [22] was used to carry out the enrichment analysis of the KEGG pathway of genes, focusing on the significant enrichment pathway with a *p*-value < 0.05.

### 2.7. Correlation Analysis of Key Gene Methylation Levels with Frozen Semen Quality and Serum Testosterone Levels

Pearson correlation analysis was used to assess the relationship between the methylation level of key genes and the cryopreserved sperm qualities or serum testosterone level. A *p*-value ≤ 0.05 denotes a significant correlation, while a *p*-value ≤ 0.01 denotes a highly significant correlation.

## 3. Results

### 3.1. Serum Testosterone Levels and Cryopreserved Sperm Qualities

The average concentration of serum T in 102 cockerels was 66.76 pmol/L. After being stored in liquid nitrogen for one week, the thawed semen from these 102 Xiaoshan Chickens was analyzed. The LIN of cryopreserved semen was 11.47%, and the total motile sperm ratio (TM) was 48.77%. Detailed information is presented in Table 1.

### 3.2. Correlation Analysis Between Cryopreserved Sperm Qualities and Testosterone Level

The correlations among various parameters of cryopreserved semen quality and serum T level in Xiaoshan Chickens were analyzed by Pearson correlation tests. The results indicate that the serum T level was highly significantly positively correlated with TM, LIN and the proportion of sperm with the normal morphology (NM) (*p* ≤ 0.01) of cryopreserved sperm, with correlation coefficients of 0.825, 0.539, and 0.481, respectively. Additionally, the serum T level was significantly negatively correlated with the percentage of sperm with distal cytoplasmic droplets (DD) and the percentage of sperm with proximal cytoplasmic droplets (PD) after freezing (*p* ≤ 0.05), with correlation coefficients of −0.429 and −0.285, respectively. LIN and TM are highly significantly positively correlated (*p* ≤ 0.01), with a correlation coefficient of 0.679. The coiled tail sperm rate (CT) shows a highly significant negative correlation with T (*p* ≤ 0.01) and a significant negative correlation with NM (*p* ≤ 0.05), with correlation coefficients of −0.385 and −0.228, respectively. Detailed results are presented in Table 2.

### 3.3. Sperm and Semen Indexes in High- and Low-Quality Groups

Out of 102 Xiaoshan Chickens, cryopreserved semen samples were ranked based on LIN, and four samples were selected for both the high-quality group (group H) and the low-quality group (group L), as detailed in Figure 1. Group H consists of samples with LIN > 25%, while group L includes samples with LIN < 5%.

Sperm parameters for group H and group L are presented in Table 3. Significant differences were observed between the two groups in LIN, TM, CT and NM (*p* < 0.05) of cryopreserved sperm. However, no significant differences were found in SM of cryopreserved sperm.

Semen parameters for group H and group L are presented in Table 4. Significant differences were observed between the two groups in the serum T level (*p* < 0.05). However, no significant differences were found between the two groups in serum levels of E_2_, SOD and GSH-px.

### 3.4. Quality and Statistics of DNA Methylation Sequencing Data

In this experiment, a total of 243.64 G of raw reads were obtained, with 219.55 G of clean bases. The GC content ranged from 23.93% to 22.64%, with an average of 23.44%. The conversion rate after bisulfite treatment was greater than 99%, with an average of 99.5%. The Q20 scores ranged from 97.57% to 97.41%, with an average of 97.51%, while the Q30 scores ranged from 94.01% to 93.7%, with an average of 93.9%. Detailed results are provided in Table 5.

### 3.5. Distribution of Genome-Wide Methylation Sites

Cytosine (C) can be categorized into three sequence types based on its context: CG, CHG, and CHH (H represents A, T, or C). As shown in Table 6, the percentage of methylated cytosine (mC/C) in cryopreserved sperm samples ranged from 2.46% to 2.79%, with an average of 2.6%. In group H, mC/C across the genome ranged from 2.46% to 2.69%, with an average of 2.54%. In group L, mC/C ranged from 2.54% to 2.79%, with an average of 2.65%. The average percentages of methylated CG (mCG/CG), methylated CHG (mCHG/CHG), and methylated CHH (mCHH/CHH) in group H were 46.3%, 0.3%, and 0.38%, respectively, while in group L, they were 47.5%, 0.3%, and 0.35%, respectively. The mCG/CG percentage is highest in both groups across the three sequence contexts, with non-mCG types constituting a minor proportion. This indicates that the predominant methylation pattern in the frozen semen of Xiaoshan Chickens is the CG type.

### 3.6. Analysis of Differentially Methylated Regions

Differentially methylated regions (DMRs) of the mCG type in two groups of cryopreserved sperm samples were identified, yielding a total of 217 DMRs. These DMRs range in length from 51 bp to 915 bp and are normally distributed, with 74.65% of them being within 200 bp (Figure 2A). Figure 2B shows the differences in DMR methylation levels of the two groups. In group H, methylation levels range from 1.34% to 95.52%, with a median of 26.73%. In group L, methylation levels range from 1.44% to 86.02%, with a median of 43.90%. Of these, 125 DMRs were located within gene body, and 16 DMRs were found in a gene promoter. These two types of DMRs are collectively referred to as differentially methylated genes (DMGs). Detailed results are provided in Table 7.

### 3.7. Enrichment and Analysis of Differentially Methylated Genes

To investigate the role of DNA methylation during sperm cryopreservation, KEGG pathway enrichment analysis was performed on DMGs. Figure 3A presents the top 30 enriched KEGG signaling pathways, with 7 pathways showing significant enrichment (*p*-value < 0.05). These pathways included the apelin signaling pathway, progesterone-mediated oocyte maturation, melanogenesis, adrenergic signaling in cardiomyocytes, gap junction, oocyte meiosis, and cell adhesion molecules. The key differential genes involved in these pathways are *ADCY2* (adenylate cyclase 2), *MYL3* (myosin, light chain 3), *KLF2* (Kruppel-like factor 2), *GNAI2* (G protein subunit alpha i2), *CPEB3* (cytoplasmic polyadenylation element binding protein 3), *TYR* (tyrosinase), *NRXN3* (neurexin 3), *NEGR1* (neuronal growth regulator 1), and *NRCAM* (neuronal cell adhesion molecule). Additionally, by consulting published studies on gene function, two DMGs potentially associated with sperm motility were identified; they were *PRKACB* (protein kinase cAMP-activated catalytic subunit beta, CABYR) and *ACSL1* (acyl-CoA synthetase long-chain family member 1). The methylation levels of *PRKACB* and *ACSL1* in group L were significantly higher than those in group H (Figure 3B,C). Specifically, the methylation level of the *ACSL1* gene was 22.0% in group H and 61.2% in group L, while the *PRKACB* gene had a methylation level of 0% in group H and 49.1% in group L.

### 3.8. Analysis of the Correlation Between Key Genes’ Methylation Levels and Cryopreserved Sperm Qualities and Serum Testosterone Level

The correlation analysis between key candidate genes *ACSL1* and *PRKACB* with semen qualities and serum testosterone level is illustrated in Figure 4. The results indicated that the methylation level of the *ACSL1* gene was highly negatively correlated with TM and LIN, with correlation coefficients of 0.819 and 0.714, respectively, and significantly positively correlated with CT and PD, with correlation coefficients of 0.376 and 0.640, respectively. There was no significant correlation between *ACSL1* gene methylation levels and T, SM, NM, or DD. The methylation level of the *PRKACB* gene was highly negatively correlated with TM, LIN and NM, with correlation coefficients of 0.824, 0.719 and 0.789, respectively, highly positively correlated with CT, with a correlation coefficient of 0.772, and significantly positively correlated with PD, with a correlation coefficient of 0.693. There was no significant correlation between *PRKACB* gene methylation levels and T, SM, or DD.

## 4. Discussion

The reproductive performance of cockerels play an important role in poultry production. This is the first study to establish a positive correlation between serum testosterone levels and the quality of cryopreserved chicken semen while also identifying differences in sperm DNA methylation levels between different cryopreservation outcomes. This study highlights *PRKACB* and *ACSL1* as key candidate genes for further investigation, providing essential insights and a theoretical framework for breeding strategies and the development of semen cryoprotectants in Xiaoshan Chickens.

The percentage of motile sperm in a straight-line trajectory is an important indicator for evaluating the quality of cryopreserved poultry sperm [18]. In this study, significant differences were observed in the percentage of motile sperm in a straight-line trajectory of Xiaoshan Chicken sperm following the same cryopreservation and thawing procedures. Further analysis revealed a significant positive correlation between the percentage of motile sperm in a straight-line trajectory in cryopreserved semen and the serum testosterone level. Research on testosterone as an indicator of sperm quality is less common in chickens, particularly regarding the quality of cryopreserved chicken semen. In this study, we established a clear relationship between serum testosterone level and the quality of cryopreserved sperm. A similar result was found in research on Beijing-You chickens, which indicates that testosterone is associated with sperm production [17]. Based on these results, we forecast that testosterone may influence sperm development and maturation, affecting sperm membrane structure, which in turn impacts their resistance to freezing. Measuring serum testosterone levels in cockerels could be a valuable tool for selecting males with semen more suitable for cryopreservation, thus improving the efficiency of frozen semen usage.

The study by Mesbah et al. [23] demonstrated that the addition of low concentrations of E_2_ to semen could improve post-thaw sperm motility in goats. However, the addition of excessively high concentrations of E_2_ to semen stimulates sperm cells, leading to oxidative stress, increased mitochondrial superoxide (O^2−^) levels, and decreased mitochondrial membrane potential [24]. Li et al. [25] found that SOD and GSH-px in porcine semen are closely related to sperm resistance to cryopreservation, helping to reduce oxidative stress caused by cryopreservation. In contrast to these findings, no significant differences were observed in the levels of E_2_, SOD, and GSH-px in the serum between the H and L groups in our study, suggesting that these factors do not significantly affect the semen cryopreservation quality in Xiaoshan Chickens.

DNA methylation is one epigenetic actor associated with sperm nucleus compaction, gene silencing, and the prepatterning of embryonic gene expression [26]. This study analyzed the differences in methylation levels between high-quality and low-quality sperm after cryopreservation, revealing that the DNA methylation level in the low-quality sperm group was higher than that in the high-quality group. This finding is inconsistent with previous studies by Liu et al. [27] and Montjean et al. [11], which examined the relationship between DNA methylation levels and sperm quality. The primary reason for this discrepancy is that both Liu and Montjean used the DNA fragmentation index as a measure of sperm quality. This suggests that the percentage of motile sperm in a straight-line trajectory and the DNA fragmentation index are two independent indicators of sperm quality.

Notably, we found that the methylation levels of the *PRKACB* and *ACSL1* genes were significantly higher in high-quality cryopreserved sperm compared to low-quality cryopreserved sperm. PRKACB protein plays an important role in the regulation of the functionality and viability of spermatozoa [28,29,30]. In this study, the methylation level of CG sites in the *PRKACB* gene was 0% in high-quality cryopreserved sperm, which was significantly lower than the 49.1% observed in low-quality sperm. Its methylation level was negatively correlated with total sperm motility, linear motility, and the percentage of normal sperm while being positively correlated with the percentage of coiled tails and proximal cytoplasmic droplets. The ACSL family, as intermediates in lipid metabolism, promotes fatty acid metabolism and membrane modification. The study by Cao et al. [31] suggests that ACSL may reduce the damage to bull sperm during cryopreservation by regulating and promoting lipid metabolism. In this study, the methylation level of CG sites in the *ACSL1* gene was 22% in high-quality cryopreserved sperm, significantly lower than the 61.2% observed in low-quality sperm. Its methylation level was negatively correlated with total sperm motility and linear motility while being positively correlated with the percentage of coiled tails and proximal cytoplasmic droplets. Our research suggests that *PRKACB* and *ACSL1* may regulate sperm qualities such as linear motility during cryopreservation by altering their methylation levels. Future research should explore how changes in methylation levels of these genes affect specific gene expression. This could reveal molecular mechanisms underlying sperm resistance to freezing.

This study integrated population-based statistics and differential group analysis to investigate the correlation between serum testosterone levels and the quality of cryopreserved semen, as well as the impact of DNA methylation on semen cryopreservation outcomes. The experimental design is feasible, and the sample size meets statistical requirements. The findings provide valuable guidance for practical applications in production: (1) measuring serum testosterone levels in cockerels could be a valuable tool for selecting males with semen more suitable for cryopreservation, thus improving the efficiency of frozen semen usage, and (2) exploring new cryoprotectants that can control DNA methylations can enhance the viability and motility of thawed chicken sperm.

## 5. Conclusions

In conclusion, the findings from this study revealed a highly significant positive correlation between the serum testosterone level and the TM, LIN and NM of cryopreserved sperms. The serum levels of E_2_, SOD and GSH-px showed no significant differences between low-quality and high-quality cryopreserved sperms. The overall methylation level of low-quality sperm was higher than that of high-quality sperm after cryopreservation. Differentially methylated genes between the two groups were primarily enriched in pathways including the apelin signaling pathway, progesterone-mediated oocyte maturation, melanogenesis, adrenergic signaling in cardiomyocytes, gap junction signaling, oocyte meiosis, and cell adhesion molecules. Notably, the methylation levels of the *PRKACB* and *ACSL1* genes were highly negatively correlated with TM and LIN in cryopreserved sperm. These findings provide both technical and theoretical references for breeding practices and the development of semen cryoprotectants for Xiaoshan Chicken.

## Figures and Tables

**Figure 1 animals-15-01745-f001:**
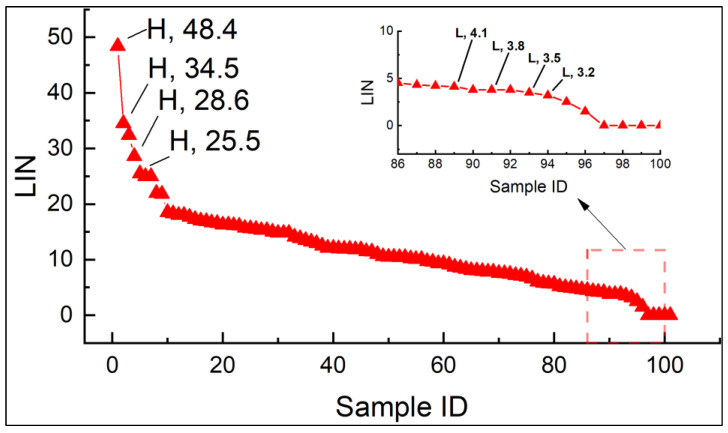
Ranking of LIN in different cryopreserved sperm samples (*N* = 102). The *x*-axis represents sample IDs, and the *y*-axis represents the percentage of motile sperm in a straight-line trajectory. H: high-quality group; L: low-quality group.

**Figure 2 animals-15-01745-f002:**
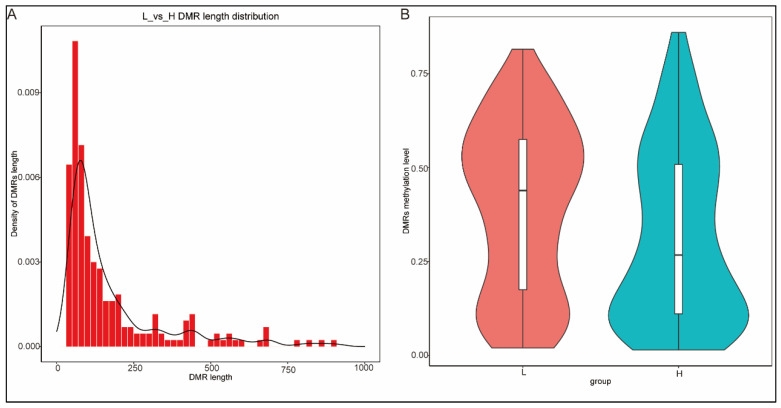
Density of DMR lengths and their methylation levels. (**A**) Density of DMR lengths; (**B**) DMR methylation levels of high-quality and low-quality cryopreserved sperm samples.

**Figure 3 animals-15-01745-f003:**
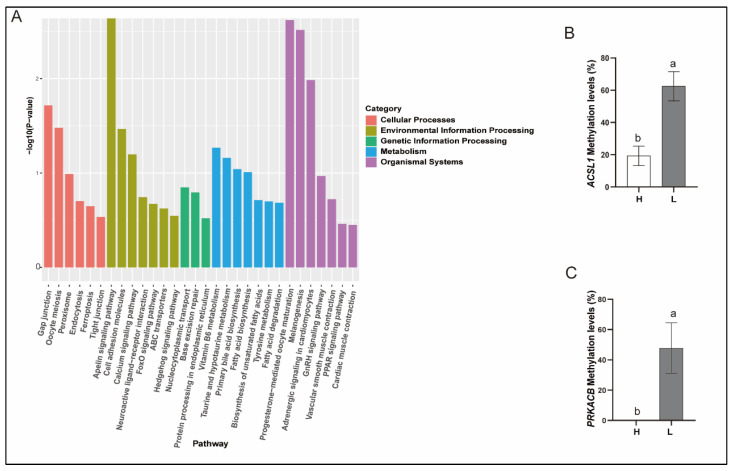
KEGG pathway enrichment of DMGs and methylation levels of key genes. (**A**) KEGG pathway enrichment of DMGs (top 30). (**B**) *ACSL1* gene methylation levels of two groups. (**C**) *PRKACB* gene methylation levels of two groups. Differing lowercase superscript letters indicate significant differences (*p* ≤ 0.05). H, high-quality group. L, low-quality group.

**Figure 4 animals-15-01745-f004:**
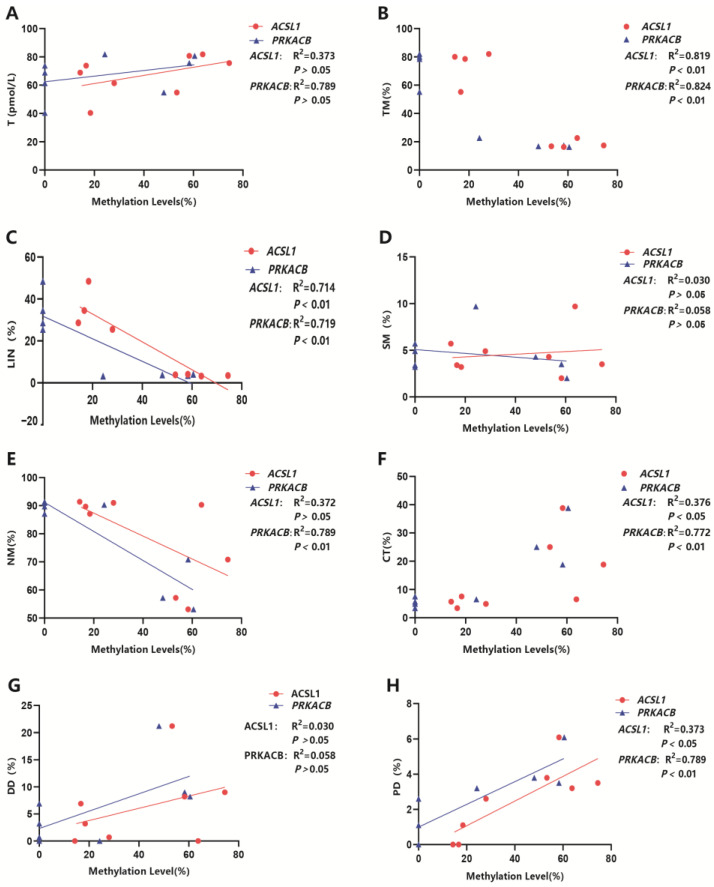
An analysis of the correlation between key genes’ methylation levels and cryopreserved sperm qualities and the serum testosterone level. (**A**) Correlation analysis between the *ACSL1* and *PRKACB* genes and serum testosterone level. (**B**) Correlation analysis between the *ACSL1* and *PRKACB* genes and the total motile sperm ratio in cryopreserved sperm. (**C**) Correlation analysis between the *ACSL1* and *PRKACB* genes and the percentage of motile sperm in a straight-line trajectory in cryopreserved sperm. (**D**) Correlation analysis between the *ACSL1* and *PRKACB* genes and the proportion of slowly motile sperm in cryopreserved sperm. (**E**) Correlation analysis between the *ACSL1* and *PRKACB* genes and the proportion of sperm with normal morphology in cryopreserved sperm. (**F**) Correlation analysis between the *ACSL1* and *PRKACB* genes and the coiled tail sperm rate in cryopreserved sperm. (**G**) Correlation analysis between the *ACSL1* and *PRKACB* genes and the percentage of sperm with distal cytoplasmic droplets in cryopreserved sperm. (**H**) Correlation analysis between the *ACSL1* and *PRKACB* genes and the percentage of sperm with proximal cytoplasmic droplets in cryopreserved sperm.

**Table 1 animals-15-01745-t001:** Serum testosterone levels and cryopreserved sperm qualities (*N* = 102).

T (pmol/L)	LIN/%	TM/%	SM/%	Dmr/%	DD/%	PD/%	NM/%
66.76 ± 17.36	11.47 ± 7.77	48.77 ± 20.25	8.05 ± 4.40	2.78 ± 2.35	4.35 ± 3.93	2.61 ± 2.31	82.40 ± 12.75

T, serum testosterone. LIN, the percentage of motile sperm in a straight-line trajectory. TM, the total motile sperm ratio. SM, the proportion of slowly motile sperm. Dmr, the percentage of sperm with bending in the distal midpiece region of tail. DD, the percentage of sperm with distal cytoplasmic droplets. PD, the percentage of sperm with proximal cytoplasmic droplets. NM, the proportion of sperm with normal morphology.

**Table 2 animals-15-01745-t002:** Correlation analysis between cryopreserved sperm qualities and testosterone level (*N* = 102).

Indexes	T	TM	LIN	SM	DD	PD	NM	CT
T	1							
TM	0.825 **	1						
LIN	0.539 **	0.697 **	1					
SM	0.15	0.283 **	−0.026	1				
DD	−0.429 **	−0.473 **	−0.245 *	−0.309 **	1			
PD	−0.285 **	−0.321 **	−0.239 *	−0.321 **	0.353 **	1		
NM	0.481 **	0.591 **	0.345 **	0.380 **	−0.493 **	−0.375 **	1	
CT	−0.385 **	−0.180	−0.121	0.014	0.128	0.086	−0.228 *	1

An asterisk (*) indicates a significant correlation (*p* ≤ 0.05), while a double asterisk (**) indicates a highly significant correlation (*p* ≤ 0.01). T, serum testosterone. LIN, the percentage of motile sperm in a straight-line trajectory. TM, the total motile sperm ratio. SM, the proportion of slowly motile sperm. DD, the percentage of sperm with distal cytoplasmic droplets. PD, the percentage of sperm with proximal cytoplasmic droplets. NM, the proportion of sperm with normal morphology. CT, the coiled tail sperm rate.

**Table 3 animals-15-01745-t003:** Comparison of cryopreserved sperm qualities between Group H and Group L (*N* = 4).

Groups	LIN/%	TM/%	SM/%	CT/%	NM/%
H	34.25 ± 10.15 ^a^	73.93 ± 12.57 ^a^	4.30 ± 1.20	0.00 ± 0.00 ^a^	89.80 ± 1.94 ^a^
L	3.65 ± 0.39 ^b^	18.28 ± 2.92 ^b^	4.88 ± 3.35	0.25 ± 0.50 ^b^	67.85 ± 16.77 ^b^

Data with different lowercase letters within the same column indicate significant differences (*p* ≤ 0.05). LIN, the percentage of motile sperm in a straight-line trajectory. TM, the total motile sperm ratio. SM, the proportion of slowly motile sperm. CT, the coiled tail sperm rate. NM, the proportion of sperm with normal morphology.

**Table 4 animals-15-01745-t004:** Comparison of cryopreserved serum indicators between Group H and Group L (*N* = 4).

Groups	T (pmol/L)	E_2_ (pmol/L)	SOD (U/mL)	GSH-px (U/mL)
H	68.01 ± 6.24 ^a^	94.50 ± 16.32	70.68 ± 6.28	2400.00 ± 96.21
L	79.36 ± 3.31 ^b^	89.07 ± 13.66	73.44 ± 10.71	2363.64 ± 218.18

Data with different lowercase letters within the same column indicate significant differences (*p* ≤ 0.05). T, serum testosterone. E_2_, serum estradiol. SOD, superoxide dismutase. GSH-px, serum glutathione peroxidase. H, high-quality group. L, low-quality group.

**Table 5 animals-15-01745-t005:** Base sequencing quality across different samples.

Sample	Reads Num	Raw Bases (bp)	Clean Bases (bp)	GC Content/%	Q20/%	Q30/%	BS Conversion Rate/%
H1	222,315,222	33,569,598,522	30,221,998,491	23.19	97.49	93.89	99.49
H2	182,670,788	27,583,288,988	24,848,377,670	23.31	97.49	93.83	99.51
H3	182,225,238	27,516,010,938	24,812,522,476	23.64	97.54	93.96	99.5
H4	193,707,308	29,249,803,508	26,349,086,697	22.96	97.48	93.83	99.5
L1	207,922,812	31,396,344,612	28,295,6320,93	23.93	97.51	93.91	99.51
L2	208,033,018	31,412,985,718	28,334,726,009	23.47	97.57	94.08	99.51
L3	206,532,318	31,186,380,018	28,066,514,129	23.79	97.41	93.7	99.5
L4	210,131,334	31,729,831,434	28,622,210,423	23.21	97.55	94.01	99.51

H1~H4 represent the four samples from the high-quality group; L1~L4 represent the four samples from the low-quality group. Reads Nmu, the total number of raw sequence data; Raw Bases, the number of Raw Reads multiplied by the length of each sequencing read; Clean Bases, the number of clean reads multiplied by the length of each sequencing read. Q20, the percentage of bases with an accuracy greater than 99%. Q30, the percentage of bases with an accuracy greater than 99.9%. BS conversion rate, the conversion rate of cytosine (C) to thymine (T) by bisulfite treatment.

**Table 6 animals-15-01745-t006:** Methylation ratio of cytosine site.

Samples	mCG/CG/%	mCHG/CHG/%	mCHH/CHH/%	mC/C/%
H1	47.30	0.30	0.40	2.52
H2	45.80	0.30	0.30	2.51
H3	47.90	0.30	0.40	2.69
H4	46.70	0.30	0.40	2.46
Mean (H)	46.93	0.30	0.38	2.54
L1	48.90	0.30	0.40	2.79
L2	46.90	0.30	0.30	2.57
L3	47.80	0.30	0.40	2.71
L4	46.40	0.30	0.30	2.54
Mean (L)	47.50	0.30	0.35	2.65
Mean (H + L)	47.21	0.30	0.36	2.60

mC/C, the percentage of methylated cytosine. mCG, mCHG, mCHH, three methylation types of cytosine based on its context.

**Table 7 animals-15-01745-t007:** Statistics of DMRs and their lengths.

DMR	Num	mCG	Length/bp
Total	217	1365	51~915
Gene	125	799	51~915
Promoter	16	128	59~135

Total, all the DMRs. Gene, DMRs located within gene body. Promoter, DMRs found in gene promoter. mCG, number of mCG sites in DMRs.

## Data Availability

The data are available from the corresponding author upon reasonable request due to privacy reasons.

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
