# Peer review of "Correlation Analysis of Sperm Cryopreservation Quality with Serum Testosterone and Sperm gDNA Methylation Levels in Xiaoshan Chickens"

_animals, 2025, doi:10.3390/ani15121745_

Round 1

Reviewer 1 Report

Comments and Suggestions for Authors

1.Line 116, please kindly describe the procedure for exposing the sample to liquid nitrogen vapor for a duration of 5 minutes.

2.Line 134-135, please provide a rationale for the division of samples into the H group and L group is evaluated solely based on straight-line trajectory. Why were alternative metrics such as TM, viability, motility, or other composite indicators not considered?

3.Line 200, "store"should be instead of "dtore".

4.Line 233 states that four samples were selected, but Tables 3 and 4 indicate a sample size of five. What is the correct sample size?

5.Could you clarify the meaning of "H1" as it appears in the final row of Table 6?

6.Table 4 shows that Group H has higher MDA levels than Group L. Could you explain this difference? The standard deviation value in the MDA was excessively high, rendering it unreliable. It is recommended to re-evaluate the original data to ensure accurate statistical analysis.

Author Response

Comments 1: Line 116, please kindly describe the procedure for exposing the sample to liquid nitrogen vapor for a duration of 5 minutes.

Response 1: Thank you for pointing out this deficiency. The straws were placed 5 cm above the liquid nitrogen and vaporized with liquid nitrogen for 5 min, and the semen was lowered from 4°C to -35°C and then immersed in a liquid nitrogen tank for long-term preservation. We have added this description in 2.1 part (Line 122-125) of new version.

Comments 2: Line 134-135, please provide a rationale for the division of samples into the H group and L group is evaluated solely based on straight-line trajectory. Why were alternative metrics such as TM, viability, motility, or other composite indicators not considered?

Response 2: Thank you for pointing out this problem. The main basis we rely on is the LIN value, as demonstrated by Bondarenko et al. (Bondarenko V, Cosson J. Structure and beating behavior of the sperm motility apparatus in aquatic animals. Theriogenology 2019;135:152–63.) have concluded that the percentage of sperm cells moving in a straight line (LIN) is the primary indicator of sperm motility. The LIN value is usually positively correlated with the TM value. In the previous version of the group description, we only provided the range of LIN values and omitted the range of TM values. In fact, when LIN is greater than 25%, TM is greater than 50%; when LIN is less than 5%, TM is less than 20%. In this version, we have made supplements in L143-145.

Comments 3: Line 200, "store"should be instead of "dtore".

Response 3: Accepted. We have made the requested changes.

Comments 4: Line 233 states that four samples were selected, but Tables 3 and 4 indicate a sample size of five. What is the correct sample size?

Response 4: Thank you for pointing out this problem. The sample size is indeed written in a very messy way in last version. Here is an explanation. We sorted all the samples according to LIN, with 4 samples each in Group H and Group L. However, when conducting the semen quality test, 5 Top samples and 5 Bottom samples were used. In the new version, we removed the added samples from Table 3 and corrected the sample sizes in Tables 3 and 4.

Comments 5: Could you clarify the meaning of "H1" as it appears in the final row of Table 6?

Response 5: Accepted. Thank you for pointing out this mistake. The final version of Table 6 has been modified through the removal of the final row (H1).

Comments 6: Table 4 shows that Group H has higher MDA levels than Group L. Could you explain this difference? The standard deviation value in the MDA was excessively high, rendering it unreliable. It is recommended to re-evaluate the original data to ensure accurate statistical analysis.

Response 6: Accepted. Thanks for the comment. Through data re-evaluation, we can not confirm the accuracy of the reported MDA results. In order to make the presented experimental results more rigorous, we removed the presentation of the MDA in Result and Discussion.

Reviewer 2 Report

Comments and Suggestions for Authors

Cryopreservation of avian sperm has not been studied much yet. Research aimed at finding markers of sperm quality after freezing is very interesting.

The manuscript is clear, relevant for the field and presented in a well-structured manner.  The cited references are mostly recent publications and relevant.  The manuscript’s results are reproducible based on the details given in the methods section.  The figures and  tables are appropriate. 

I have a few comments.

  1. At what temperature did you keep the straws in the nitrogen vapor?
  2. I would like to see a table with sperm quality data before freezing.
  3. Why  the morphology and  the membranes integrity were not assessed, but only sperm motility assessed?
  4. Complete the conclusion

Author Response

Comments 1: At what temperature did you keep the straws in the nitrogen vapor?

Response 1: Thank you for pointing out this problem. The straws were placed 5 cm above the liquid nitrogen and vaporized with liquid nitrogen for 5 min, and the semen was lowered from 4°C to -35°C and then immersed in a liquid nitrogen tank for long-term preservation. We have added this description in 2.1 part (Line 122-125) of new version.

Comments 2: I would like to see a table with sperm quality data before freezing.

Response 2: Thanks for your advice. Due to the sampling environment limitations, we were unable to carry computer-assisted sperm analysis (CASA) during sperm collection to assess sperm quality prior to cryopreservation. However, we evaluated the fresh semen through an ordinary optical microscope. To ensure quality and motility, only fresh semen samples with motility greater than 70 % and sperm concentration of approximately 200-300 million/mL were used for cryopreservation. We have added this information in 2.1 Sampling.

The main research point was the difference between the high and low motility sperm groups after freezing, while the control group was already set up to meet the requirements. And the control group set up already meets the requirements. On the other hand, we also think the reviewers' suggestion is excellent, but because of some reasons the sperm qualities before freezing were not recorded, and it is very regrettable that we can't provide them.

Comments 3: Why the morphology and the membranes integrity were not assessed, but only sperm motility assessed?

Response 3: Thank you for pointing out this problem. Sperm motility is an important indicator for assessing the quality of poultry sperm cryopreservation, which directly affects the fertilization rate, so we chose sperm motility as an evaluation indicator. We used computer-assisted sperm analysis (CASA) to evaluate the quality of spermatozoa, and the morphological indicators such as Distal Midpiece Reflex (Dmr), Distal Droplet (DD), Proximal Droplet (PD), Normal Morphology (NM), were measured, but no pictures were saved. Membrane integrity is another important indicator for evaluating sperm quality, In the experimental design, we believed that the indicators obtained by the CASA instrument were sufficient to reflect the sperm motility. However, we overlooked this point, and we feel very regretful for this.

Comments 4: Complete the conclusion

Response 4: Thanks for your advice. We completed the conclusion by:

(1) added the correlation study part of serum testosterone (complete the correlation index).

(2) added the summary of serum indicators study.

(3) added the specific indicators of cryopreserved sperm that were negatively correlated with the methylation levels of the PRKACB and ACSL1 gene.

Reviewer 3 Report

Comments and Suggestions for Authors

The manuscript "Correlation Analysis of Sperm Cryopreservation Quality with Serum Testosterone and Sperm gDNA Methylation Levels in Xiaoshan Chickens" describes the covariation of different parameters characterizing chicken sperm after freeze-thaw cycle. Testosteron was found to be the best predictor of semen quality, while the DNA methylation was shown to be higher in low-quality semen than in high-quality.

Comments:

  1. It seems from Fig. 1 that the frequency distribution of quality parameters is not normal with the heavier tail on low-quality. Is this so? If so, why?
  2. For some pairs of parameters, even if statistical significance level is reached, the correlation coefficients are so low that the linear relationship is not adequate (SM and TM; PD and T and so on).
  3. It would be interesting to see the quaility control of sperm before freezing cycle. How hazardous for semen is the cryopreservation?

Typos found: line 199, "dtore"

Author Response

Comments 1: It seems from Fig. 1 that the frequency distribution of quality parameters is not normal with the heavier tail on low-quality. Is this so? If so, why?

Response 1: Figure 1 showed the LIN of each sample. Here we made the frequency distribution figure  in atteched file. It demonstrated that the frequency distribution of quality parameters is normal.

Comments 2: For some pairs of parameters, even if statistical significance level is reached, the correlation coefficients are so low that the linear relationship is not adequate (SM and TM; PD and T and so on).

Response 2: Thanks for the comment. The correlation coefficient between SM and TM was 0.283, the correlation coefficient between PD and T was -0.285. The P values of them were ≤ 0.01. These indicated week correlations between SM and TM, and PD and T. The correlation implies that there is more to understand and that there are extraneous or underlying factors that should be explored further in order to search for a cause. Other weak correlations also apply to this explanation.

Comments 3: It would be interesting to see the quaility control of sperm before freezing cycle. How hazardous for semen is the cryopreservation?

Response 3: Thanks for the comment. Due to the sampling environment limitations, we were unable to carry computer-assisted sperm analysis (CASA) during sperm collection to assess sperm quality prior to cryopreservation. However, we evaluated the fresh semen through an ordinary optical microscope. To ensure quality and motility, only fresh semen samples with motility greater than 70 % and sperm concentration of approximately 200-300 million/mL were used

for cryopreservation. We have added this information in 2.1 Sampling.

Cryopreservation causes great damage to semen. In reports related to the cryopreservation of chicken semen, most of the thawing motility typically ranges between 50% and 70% (Zong Y., Li Y., Sun Y., et al. 2023. Chicken sperm cryopreservation: review of techniques, freezing damage, and freezability mechanisms[J]. Agriculture. 13(2): 445.), varying depending on breed and methods. For instance, the post-thaw motility of Ross chickens under different cryopreservation conditions ranges from 55.1 % to 74.4 % (Masoudi, R., Sharafi, M., Shahneh, A.Z. , et al. 2018. Supplementation of extender with coenzyme Q10 improves the function and fertility potential of rooster spermatozoa after cryopreservation. Anim. Reprod. Sci. 198, 193–201.; Mehdipour, M., Daghigh Kia, H., Najafi, A., et al., 2020. Effect of crocin and naringenin supplementation in cryopreservation medium on post-thaw rooster sperm quality and expression of apoptosis associated genes. PLoS One 15 (10), e0241105.; Najafi, A., Kia, H.D., Mehdipour, M., et al., 2020. Effect of quercetin loaded liposomes or nanostructured lipid carrier (NLC) on post-thawed sperm quality and fertility of rooster sperm. Theriogenology 152, 122–128.; Najafi, A., Daghigh-Kia, H., Mehdipour, M., Mohammadi, H., et al. 2022. Comparing the effect of rooster semen extender supplemented with gamma-oryzanol and its nano form on post-thaw sperm quality and fertility. Poult. Sci. 101 (3), 101637.). Another study indicated that after a certain cryopreservation and thawing process, the motility of Hy-Line White chickens was only 24.2% (Mosca, F., Zaniboni, L., Sayed, A.A., et al., 2019. Effect of dimethylacetamide and N-methylacetamide on the quality and fertility of frozen/ thawed chicken semen. Poult. Sci. 98 (11), 6071–6077.).

Comments 4: Typos found: line 199, "dtore"

Response 4: Accepted. We have corrected it in part 3.1 (Line 208) of new version. 

Round 2

Reviewer 3 Report

Comments and Suggestions for Authors

The revised manuscript is suitable for publication as the corrections made and answers provided are satisfactory.